

# Confirmation of age-related alterations in inhibitory control using a modified minimally delayed oculomotor response (MDOR) task

Paul C. Knox[1] and Dongmei Liang[1,2]

[1] Department of Eye and Vision Science, Institute of Life Course and Medical Sciences, University of Liverpool, Liverpool, United Kingdom
[2] School of Physical Education & Sports Science, National Demonstration Centre for Experimental Sports Science Education, South China Normal University, Guangzhou, China

Corresponding author
Paul C. Knox, pcknox@liv.ac.uk

## ABSTRACT

Considerable effort has been made to measure and understand the effects of ageing on inhibitory control using a range of behavioural tasks. In the minimally delayed oculomotor response (MDOR) task, participants are presented with a simple visual target step with variable target display duration (TDD), and instructed to saccade to the target not when it appears (a prosaccade response), but when it disappears (i.e., on target offset). Using this task, we recently found higher error rates and longer latencies for correct responses in older compared to younger participants. Here we have used a modified MDOR task, in which participants were presented with static placeholders identifying potential target positions (increasing spatial information), and three TDDs rather than two (reducing temporal predictability). We found that the yield of analysable trials was generally higher with this modified task and in 28 older (mean ± SD age: 65 ± 7 y) and 25 younger (26 ± 7 y) participants the total overall error rate was again higher in the older group (30 ± 18% vs. 16 ± 11%). An analysis of the temporal distribution of responses demonstrated a pronounced peak in error production around 150 ms (young) or 200 ms (old) after target onset. When we recalculated the error rate focusing on these errors, it was again significantly higher in the older group. The latency of correct responses (to offsets) was significantly increased in the older group, although much of this increase was accounted for by expected age-related visuomotor slowing. However, both latency and distribution data suggested that while older participants could generate increased levels of inhibition, they could not maintain these levels as efficiently as the younger participants. In 24 participants (15 old, 9 young) who completed both versions of the MDOR task, neither latency nor error rates differed significantly between versions. These results confirm an inhibitory control deficit in healthy older participants, and suggest that the dynamics of inhibitory control are also affected by ageing. The modified MDOR task yields more data while not altering basic performance parameters.

## INTRODUCTION

Stopping ourselves from executing motor actions or thinking thoughts is at least as important as being able to execute those actions or generate those thoughts. Without some means of inhibitory control, both behaviour and cognition become disordered. For this reason, the phenomenon of inhibitory control has been an important subject of investigation (*Bari & Robbins, 2013*). Given that actions, unlike thoughts, lend themselves to third party observation and measurement, various tasks have been developed to study behavioural inhibitory control with the conviction that these provide insights into underlying cognitive processes (*Castiglione et al., 2019*; *Logan & Cowan, 1984*; *Logan, Schachar & Tannock, 1997*). A number of oculomotor tasks have been used to study inhibitory control, including ocular no go (*Crawford et al., 2005*), countermanding (*Hanes & Carpenter, 1999*) and antisaccade (AS) tasks (*Alichniewicz et al., 2013*; *Hallett, 1978*). To this list we recently added the minimally delayed oculomotor response (MDOR) task (*Knox & Pasunuru, 2020*; *Knox, Heming De-Allie & Wolohan, 2018*; *Wolohan & Knox, 2014*).

The MDOR task is a saccade task in which the stimulus is a target step with randomised direction and timing. Participants execute a saccade to a target to the left or right of fixation, not when it appears but when it is extinguished. Target display duration (TDD) is varied so that the onset of the target does not reliably predict the time of the offset. Performance is measured using the error rate (the proportion of trials in which participants execute a target directed saccade prior to target offset) and the latency of correct responses. While the MDOR task shares features with both AS and memory-guided saccade tasks (MGS; see *Hutton (2008)*, for a review), it differs from both in important ways. As in the AS task, the error rate provides information about inhibitory control performance. However, in the AS task there is a competition between processes which in each trial leads to either an error prosaccade or a correct antisaccade (*Munoz & Everling, 2004*). Because the AS error rate is the product of this race between two processes competing for behavioural expression, an error prosaccade is not necessarily the result of a failure of inhibition: the error rate, while related to the effectiveness of inhibitory control, cannot provide an unambiguous measurement of that control (*Reuter et al., 2007*; *Reuter, Rakusan & Kathmanna, 2005*). In the MDOR task there is no such competition. Further, in the MDOR task by identifying responses to target *onsets*, which by definition must be inhibition failures, an error rate specifically and closely related to the effectiveness of inhibition can be calculated.

We assume that there is a working memory load in the MDOR task; at a minimum there is an instruction to be remembered. But any memory load is reduced compared to both AS and MGS tasks, because it is not necessary to remember a target location; the target is present throughout the period of central fixation prior to target offset. Visuospatial attention is involved in all types of saccade tasks (*Kowler et al., 1995*) with a close link between the allocation of attention and saccade programming (*Deubel, 2008*). In the AS task attentional resources are divided or distributed between alternative target locations (*Klapetek, Jonikaitis & Deubel, 2016*). In most versions of the MGS task, the disappearance

of a central fixation target provides the go signal for a saccade directed to the remembered target position, therefore involving two distinct locations and implying some division of attentional resources. In the MDOR task, it is the offset at the target location that provides both the go signal and indicates the position to which the saccade should be directed. Taken together, these considerations suggest that the MDOR task provides a means of investigating behavioural inhibition relatively uncontaminated by those attentional and memory processes that, along with inhibitory control, are key components of executive function (*Friedman & Miyake, 2017*; *Miyake et al., 2000*).

We recently used the MDOR task to investigate the effect of normal healthy ageing on behavioural inhibitory control (*Knox & Pasunuru, 2020*). Comparing a group of older participants with previously collected data from younger participants, we found that error rates were higher and the latency of correct responses longer in the older group. Further investigation of the latency differences suggested that most of the increase in latency was accounted for by general age-related slowing in the visuomotor system (*Salthouse, 1996*; *Verhaeghen, 2011*).

However, in the task used in that study, the stimulus display was sparse, with only simple fixation and saccade targets presented. At the end of each trial, after the eccentric target was extinguished (providing both the go signal and the target location for the saccade), the stimulus monitor was completely blank. This led to a number of potential uncertainties on the part of participants. There was no visual feedback as to the spatial accuracy of their saccade, no marker for how long they should fixate the eccentric position and no formal marker of the end of the trial. This, in turn, could lead to early blinks and breaks of fixation, and the loss of data. We also only used two TDDs (200 ms and 1,000 ms) limiting the inferences that could be made with regard to the timing of errors and differences in timing and rates between errors and correct responses. And we were comparing the data collected from older participants with historical data collected in earlier experiments.

We have addressed these and other issues in the current experiment by the addition of static placeholders (at the potential positions of the saccade targets) and the use of three rather than two TDDs. The placeholders reduce the spatial uncertainty in the task; participants have increased information about *where* to saccade to. The appearance and disappearance of the placeholders also provides a clear indication of the start and end of trials, delineating the intertrial period. The introduction of a third, intermediate TDD (600 ms), reduces the temporal predictability of the task, while also proving additional data about the temporal distribution of responses.

In this new experiment, both old and young participants (returning and naïve) have been contemporaneously tested using the modified MDOR task, in order to further investigate task performance and ageing effects reported previously. Retesting previous participants with the new version of the MDOR task (as well as recruiting and testing naïve participants) allowed an investigation of general laboratory or task familiarity effects, as well as an explicit comparison of the cumulative effects of the changes made in the new task, all issues that relate to performance stability.

## MATERIALS & METHODS

### Ethics and participants

Healthy, adult participants, with normal or corrected to normal vision were recruited, under ethical approval from the University of Liverpool Research Ethics committee, Liverpool, England (Reference number: 2933) and the study was conducted in accord with the ethical standards laid down in the 1964 Declaration of Helsinki. Interested potential participants were provided with a study information sheet. Having read this, and after the experiment was explained to them and they had an opportunity to ask questions, written consent was obtained. Older participants were offered £10 to compensate them for their time and for the expense of travelling in to the University for testing. Younger participants, who were recruited from the University, were offered £5.

### Apparatus and stimuli

We used the same apparatus as in previous MDOR experiments (*Knox & Pasunuru, 2020*; *Knox, Heming De-Allie & Wolohan, 2018*). Briefly, stimuli were presented on a 21″ monitor (1,024 × 768 spatial resolution, 100 Hz temporal resolution) driven by a VSG2/5 card (Cambridge Research Systems, Rochester, UK), positioned on the fronto-parallel plane 57 cm from the participant's eye. Horizontal eye position of the left eye was recorded using a Skalar Iris IR Eye Tracker, with the eye tracker output digitised at 1 kHz with 16-bit precision using a CED Power 1401 (Cambridge Electronic Design, Cambridge, UK) interface. Oculomotor data were stored for off-line, trial-by-trial analysis using custom software.

Figure 1 illustrates the modified MDOR task used in the present study. As noted in the Introduction, there are two key differences between this version of the task and the version used previously (see Fig. 1A in *Knox & Pasunuru (2020)*, for comparison): the inclusion of two target placeholders to the left and right of fixation (within which the saccade target appeared), and the use of three, not two, target display durations (TDD). Each trial began with the appearance of a central fixation target (a 0.2° black square on a light background) and the two target placeholders. The place holders were themselves 0.8° square boxes centred 5° from the fixation target. The fixation target was presented for a randomised period of 0.5 to 1.5 s. A synchronous MDOR task was used in which, when the fixation target was extinguished, the saccade target appeared at the centre of one of the placeholders (5° from the fixation target) and was displayed for 200, 600 or 1,000 ms. The offset of the target was the go signal for the saccade.

### Procedures

At the beginning of each testing session, the task was explained to the participant by stepping through it, before a number of trials were run at actual speed. Participants were then carefully positioned by adjusting table height, chin rest and cheek pads, and the eye tracker was applied and adjusted as necessary. Participants were instructed to maintain fixation centrally, until the target in one of the placeholders was extinguished. Then they were to saccade to the centre of the empty placeholder previously occupied by the target, and to maintain fixation there until both placeholders were extinguished (after 1 s),

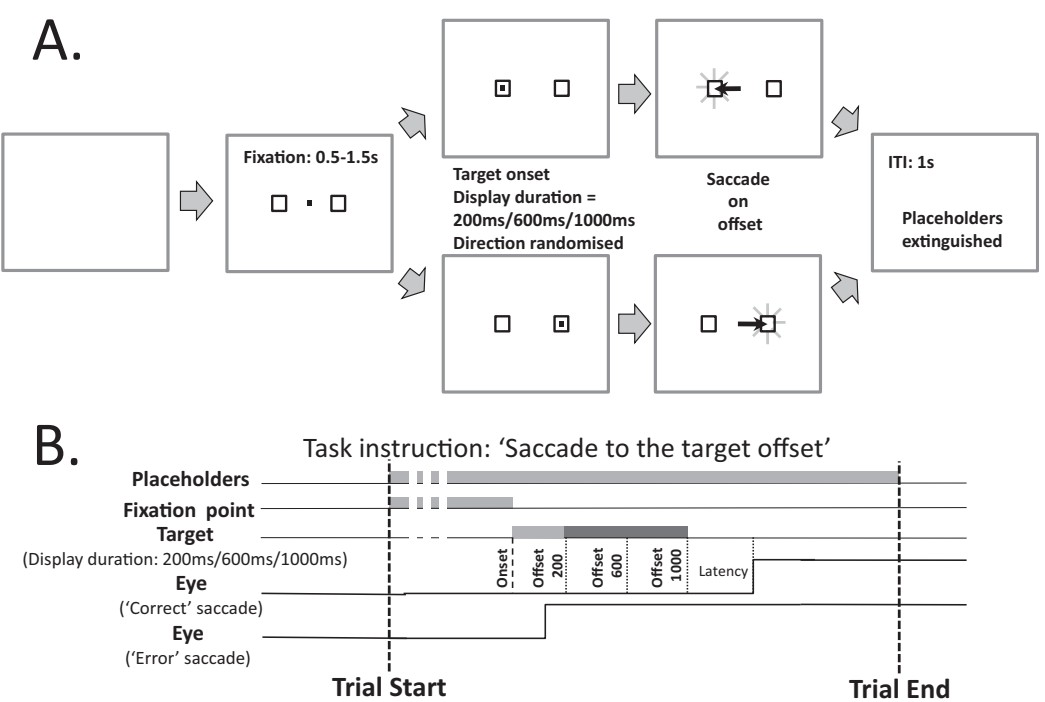

**Figure 1 Schematic description of the modified MDOR task used in this study.** (A) Each trial began with the appearance of the central fixation target, and two 0.8° × 0.8° placeholders, centred 5° to the left and right. After a random period of 0.5–1.5 s the fixation target was extinguished and the target for the saccade appeared randomly at the centre of either the left or right placeholder, and was displayed for 200, 600 or 1,000 ms (randomised). The extinction of the target was the signal for the participant to saccade to the centre of the placeholder in which the target had appeared, and maintain fixation there. After 1 s the placeholders were extinguished, at which point the participant was instructed to return fixation to the centre of the display and await the beginning of the next trial. (B) The relative timing of the various events in each trial. Note that the eye trace illustrating a correct response assumes a target display duration of 1,000 ms.

leaving the monitor blank. It remained blank until the beginning of the next trial. Participants were asked to refrain from blinking except during the intertrial interval when the monitor was blank. Practice trials were run; these data were not retained. Each participant then completed two runs of 150 MDOR trials (in the original version of the task the run length was 120 trials) with a break in between them. The quality of performance was carefully monitored to ensure that it was maintained, with verbal feedback given as necessary.

In order to calibrate eye tracker output, at the end of each MDOR run, a 32-trial calibration procedure was performed (again the same procedure as used previously in *Knox & Pasunuru (2020)* and *Knox, Heming De-Allie & Wolohan (2018)*. Calibration trials were simple prosaccade trials in which, after a randomised fixation period (0.5–1.5 s), the fixation target was extinguished and a saccade target was presented to the left or right at an eccentricity of 5° or 10° (randomised and with equal frequency) for 1 s. Participants were instructed to fixate the central point and saccade to the target as soon as it appeared, fixating it until it was extinguished, at which point they could blink and return to the

centre, ready for the next trial. All older participants also completed the Addenbrookes Cognitive Examination (ACE) III questionnaire (*Hsieh et al., 2013*).

## Analysis

Oculomotor data were analysed, as previously (*Knox & Pasunuru, 2020*; *Knox, Heming De-Allie & Wolohan, 2018*), using an interactive program which displayed the eye position data and the time at which the "go" signal (target offset in MDOR trials) occurred. The calibration data were used to transform the data from arbitrary system units into units of degrees of eye rotation. Trials with blinks or unstable fixation prior to target appearance were removed from the analysis. For each valid MDOR trial, the latency (the time from target offset to saccade onset) and amplitude of the primary target-directed saccade were measured.

Data were collated in MS Excel. Any target directed saccade with an amplitude greater than $1°$ that occurred from 80 ms after target onset to 80 ms after target offset was counted as an error. Error responses were removed and collated separately from correct trials, and the error rate calculated. Any target directed saccade occurring from 80 to 1,000 ms after target offset was counted as a correct response. For each participant, median correct saccade latency was calculated along with the error rate for each TDD. We also calculated the overall total error rate collapsing across TDDs and subsequently derived other error rates, as described in the "Results". Across participant groups, latency and error rate were summarised using the mean. In order to obtain an estimate of prosaccade latency, we calculated median saccade latency from each participant's calibration runs, collapsed across direction, eccentricity and run.

Statistical analysis was conducted with SPSS v25. Repeated measures ANOVA was used to compare groups and conditions with $\eta_p^2$ reported for effect size (details in Results). Cohen's $d_s$ (*Lakens, 2013*) is also reported where relevant.

## RESULTS

### Participant characteristics

A total of 53 participants were recruited and tested in this study. Of 28 older participants (17 male; mean ± SD age: 65 ± 7 y; the "old" group), 16 were naïve to testing and 12 had taken part in our previous study on ageing. The mean ACE III total score for the older group of 28 participants was 95 (range 89–100). The cut-off for suspicion of neurological disease is a score of 88 (*Noone, 2015*) and the mean score for our group was very similar to that reported in *Hsieh et al. (2013)* for a similarly aged healthy control group. Of 25 younger participants (mean age: 26 ± 7 y; the "young" group), 21 were naïve to testing and four had participated in a previous experiment. A total of 15,075 trials were available for analysis (7,828 from the old group, 7,247 from the young group). The mean trial yield per participant was 93 ± 8% and 97 ± 4% for the old and young groups respectively (t = 1.98, df = 51, $p = 0.05$).

### Analysis of saccade latency

The latency of correct MDOR responses across conditions followed the same pattern reported previously (Fig. 2). Note that one participant in the old group exhibited extremely

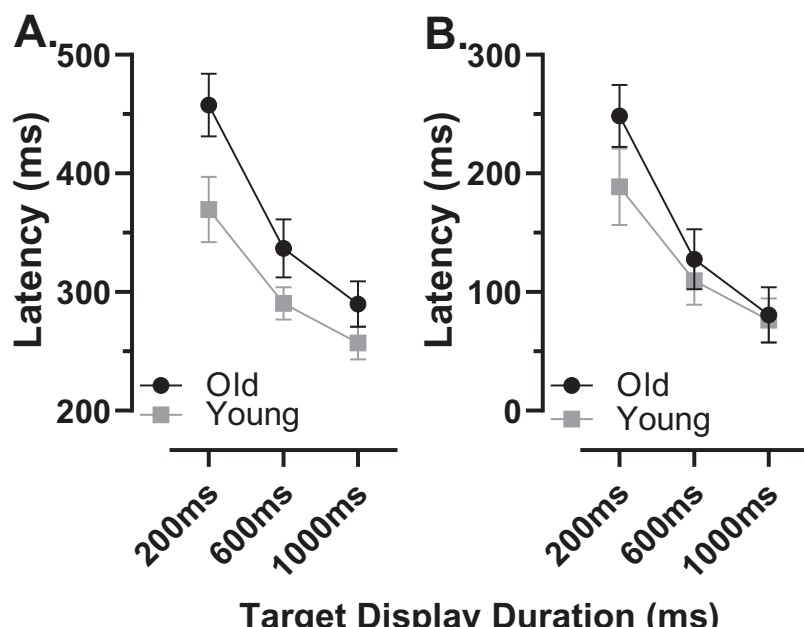

**Figure 2 Latency of correct MDOR responses.** Mean (±95% CI) latency is plotted for each target display duration for old and young groups. (A) Uncorrected latency. (B) Corrected latency; each participant's calibration task latency was subtracted from their MDOR latency, and group mean latency calculated.

high error rates for two of the TDDs leaving fewer than five correct trials from which to calculate latency. We therefore removed this participant's data from the latency analysis (leaving $n$ = 27). In both participant groups, raw latency (Fig. 2A) was longest in those trials with the shortest TDD (200 ms mean ± SD; old: 458 ± 67 ms, young 370 ± 67 ms) and was clearly modulated by TDD. For both groups there was a monotonic relationship between TDD and latency. For the longest TDD (1,000 ms) latency was 290 ± 48 ms and 257 ± 34 ms in the old and young groups respectively. It was consistently longer in the older group by 88, 47 and 33 ms for TDDs of 200, 600 and 1,000 ms respectively. These latency data were tested with repeated measures ANOVA, treating TDD as within and group as between subjects factors. Both TDD ($F_{2,49}$ = 160, $p$ < 0.001, $\eta_p^2$ = 0.87) and group ($F_{1,50}$ = 18, $p$ < 0.001, $\eta_p^2$ = 0.26) returned statistically significant results, with a statistically significant interaction between factors ($F_{2,49}$ = 6.4, $p$ = 0.003, $\eta_p^2$ = 0.21). The difference between groups for each TDD was statistically significant (t-test with Holm–Sidak correction for multiple comparisons; $p$ < 0.01 for all pairs) with effect sizes (Cohen's $d_s$) of 1.17, 0.94 and 0.78 for TDDs of 200, 600 and 1,000 ms respectively.

When we compared the latency of reflexive prosaccades in old and young groups (using data from the calibration task), latency in the old group (209 ± 44 ms) was significantly longer compared to the young group (181 ± 41m s; t = 2.4, df = 50, $p$ = 0.02; $d_s$ = 0.66). Given that this difference represents the general age-related slowing expected for the visuomotor system, we investigated whether once this was taken into account, MDOR latency still differed between groups. We calculated "corrected" MDOR latency values for each participant by subtracting their median calibration latency from the MDOR latency,

and then recalculated MDOR group means (Fig. 2B). This had the effect of reducing the latency difference between groups to 60, 18 and 4 ms for TDDs of 200, 600 and 1,000 ms respectively. When ANOVA was run on these data, the effect of TDD was unchanged ($F_{2,49}$ = 160, $p < 0.001$, $\eta_p^2$ = 0.87), the difference between groups no longer reached statistical significance ($F_{1,50}$ = 3.2, $p$ = 0.081, $\eta_p^2$ = 0.06) but there remained a statistically significant interaction between the factors ($F_{2,49}$ = 6.4, $p$ = 0.003, $\eta_p^2$ = 0.21). Only the difference at 200 ms was statistically significant (t = 2.96, df = 50, adjusted $p$ = 0.01) and $D_s$ was now 0.83, 0.31 and 0.10 for TDDs of 200, 600 and 1,000 ms respectively.

In order to quantify and investigate the modulation of latency by TDD, taking all of the data into account, we calculated the least-squares linear regression of latency on TDD for each participant. For both groups this captured the relationship reasonably well, given that the mean $r^2$ for these functions was 0.89 ± 0.12 and 0.86 ± 0.12 for old and young groups respectively. Using the regression slopes to represent the overall modulation in latency, we found that the mean slope for the old group was −0.21 ± 0.08 and was −0.14 ± 0.07 for the young group; this difference between slopes was statistically significant (t = 3.5, df = 50, $p < 0.001$, $d_s$ = 0.99). To provide comparative data with values reported previously (*Knox & Pasunuru, 2020*) we also calculated the difference in latency between the 200 and 1,000 ms TDD for each participant; the group means for this difference were 168 ± 61 ms and 113 ± 52 ms for old and young groups respectively (t = 3.5, df = 50, $p$ = 0.001, $d_s$ = 0.97).

## Analysis of errors

The overall error rate, for all errors calculated across TDD conditions, was considerably higher in the old compared with the young group (30 ± 18% vs. 16 ± 11%; t = 3.3, df = 44, $p$ = 0.002; $d_s$ = 0.93). When calculated for each TDD, the total error rate ($ER_t$) was higher in the older group and there was a modulation with TDD (it increased with TDD; Fig. 3). When analysed with a similar ANOVA design as used above for latency, TTD returned a statistically significant result ($F_{2,50}$ = 94, $p < 0.001$, $\eta_p^2$ = 0.79) as did group ($F_{1,51}$ = 10, $p$ = 0.002, $\eta_p^2$ = 0.17), with a statistically significant interaction between the factors ($F_{2,50}$ = 7.2, $p$ = 0.002, $\eta_p^2$ = 0.22). The group difference in $ER_t$ for each TDD was statistically significant (adjusted $p$ values 200 ms:$p$ = 0.03, 600 and 1,000 ms:$p < 0.01$), and $d_s$ for each TDD was 200 ms:0.67, 600 ms:0.99 and 1,000 ms:0.81.

The $ER_t$ captured all the errors as defined in the "Methods". We investigated the pattern of errors further by constructing average distributions of all responses for the two participant groups (Fig. 4). In each of the six distributions in Fig. 4 there are two clear peaks, identified by the arrows. The larger of these is composed of the correct responses, which occur after the target offset (the vertical dashed line at 0 ms in each plot). The bins prior to +80 ms contain error responses, and errors (all of which contribute to the $ER_t$) occur throughout the period between target onset (at −200, −600 and −1,000 ms) and target offset. Errors are not distributed uniformly throughout this period, and the second clear feature in these distributions is the early peak just after target onset. The timing of this peak is consistent with these responses being primarily uninhibited responses to the target onset.

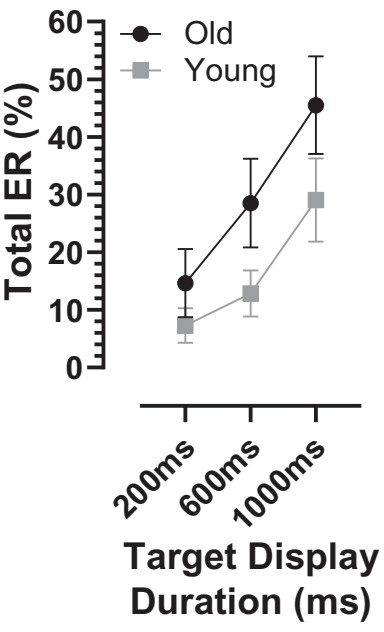

**Figure 3 Total error rate.** Mean (±95% CI) total error rate (%) is plotted for each target display duration for old and young groups.

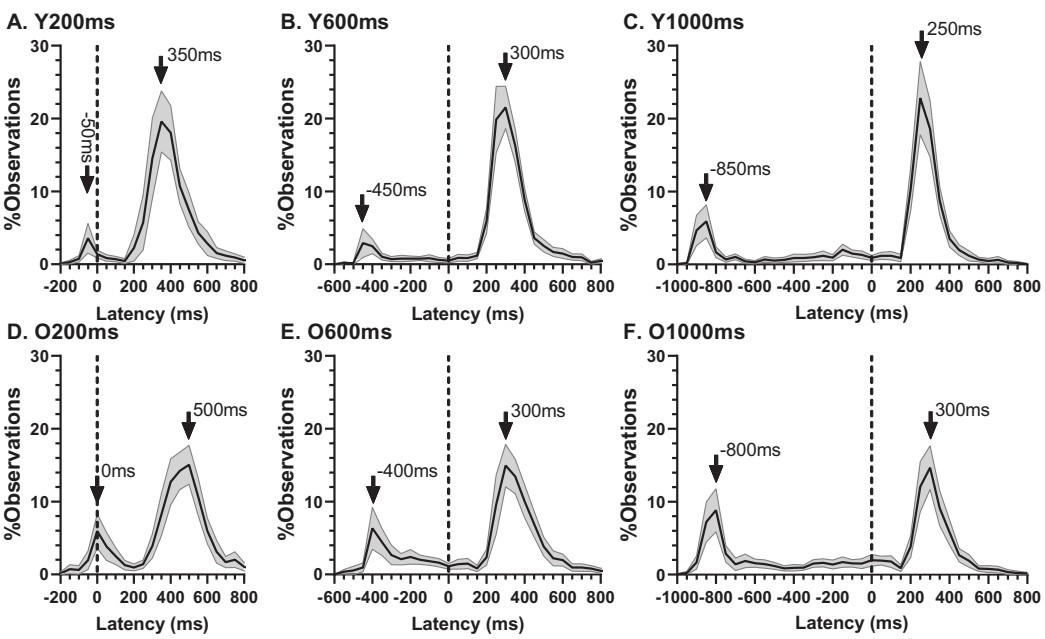

**Figure 4 Average distribution plots for young (A, B, C) and old (D, E, F) participant groups.** Bin width = 50 ms. For each participant, the % frequency distribution for all responses was calculated. The mean (±95% CI) was then calculated for each bin across participants in each group. The black central line is plotted through the each mean bin value, while the grey range shows ±95% CI. The data were not smoothed. Target onset is at −200 for 200 ms TDD (A, D), −600 for 600 ms TDD (B, E) and −1,000 ms for 1,000 ms TDD (C, F); target offset (the go signal) is at 0 ms (vertical dotted line). Arrows mark peak bins in each distribution for both errors and correct responses, and the timing of that bin is also shown.

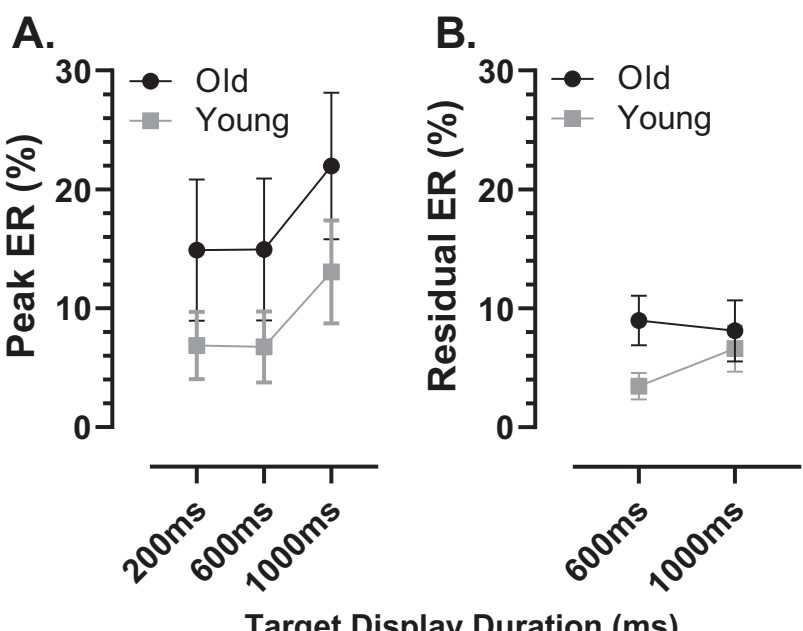

**Figure 5 Peak and residual error rates.** Mean (±95% CI) MDOR peak error (A) and residual error rates (B) in old and young participant groups. The derivation of these error rates is described in the text.

Using the average distributions, we defined two further error rates. To capture the early peak, we calculated the proportion of saccades contained in the five histogram bins centred on the peak bin as indicated in Fig. 4 (i.e., covering a latency range of 250 ms). This was calculated for each participant individually, and group means then calculated. We will refer to this as the peak error rate ($ER_{pk}$). As can be seen from Fig. 5A (and consistent with Fig. 4), the $ER_{pk}$ was consistently higher in the older group although no longer as clearly modulated by TDD. When tested with ANOVA, the TDD did have a significant effect on the rate ($F_{2,50} = 43$, $p < 0.001$, $\eta_p^2 = 0.63$) as did group ($F_{1,51} = 6.1$, $p < 0.017$, $\eta_p^2 = 0.12$); the TDD x Group interaction was not statistically significant ($F_{1,51} = 0.2$, $p = 0.8$). For the 600 and 1,000 ms TDDs, we also defined a "residual" error rate ($ER_r$). As can been seen from Fig. 4, errors continued to occur at a low rate after the peak. While for the shortest TDD (200 ms) it was difficult to distinguish between the end of the initial peak and the beginning of correct responses to the target offset, this was not the case for the longer TDDs. We therefore used the five bins (the same number of bins used for the $ER_{pk}$ rate) from −250 to 0 ms to calculate $ER_r$ for the two longer TDDs (Fig. 5B), and calculated the proportion of errors contained in these bins for each participant. The difference between groups was now less clear, partly because the average number of errors was relatively low, particularly in the young participants. Statistically, there was now no significant effect of TDD ($F_{1,51} = 2.3$, $p = 0.133$), although both the interaction ($F_{1,51} = 7.2$, $p = 0.01$, $\eta_p^2 = 0.12$) and the group effect were significant ($F_{1,51} = 8.7$, $p = 0.005$, $\eta_p^2 = 0.14$).

As indicated by the arrows on Fig. 4, the early peaks occurred slightly later in time in the old compared to the young group. To examine the timing of these early error saccades

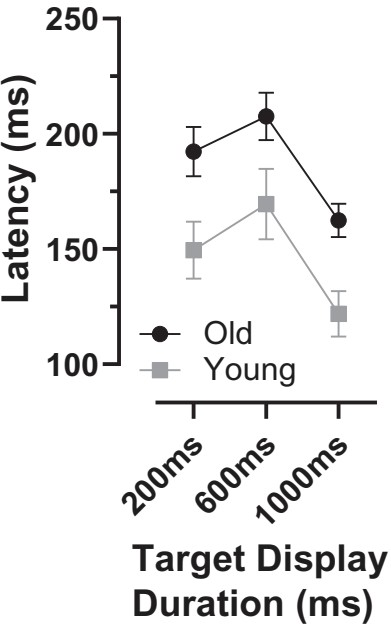

**Figure 6 Latency of "peak" errors.** Mean (±95% CI) latency of peak errors for old and young participant groups. Details of how these were calculated are in the text.

further, we used the bin ranges discussed above (five bins centred on the peak bin in each distribution, a range of 250 ms), calculated the median latency of saccades falling in these ranges for each participant and then calculated and compared group means (Fig. 6). Saccade latency was longer in the old group as would be expected given the general age-related slowing in the old group noted above. The latency difference was 43, 29 and 30 ms for the TDDs of 200, 600 and 1,000 ms respectively. When tested with a repeated measures ANOVA similar to those described previously, the difference between groups was statistically significant (Group: $F_{1,47} = 51$, $p < 0.001$, $\eta_p^2 = 0.52$; TDD: $F_{2,46} = 94$, $p < 0.001$, $\eta_p^2 = 0.80$; interaction: $F_{2,46} = 0.4$, $p = 0.68$). Note that there were two young and one old participant in whom no median latency was available for at least one TDD; data from these participants were not included in the ANOVA.

We also investigated whether there was any relationship between each participants' prosaccade latency and the latency of the error saccades comprising the early peak (Fig. 7). While the general context in which these prosaccades and MDOR peak error saccades were executed was very different, given that in both cases these were reflexive responses to target onsets, it seemed plausible that there might be a relationship. Overall correlations (in which old and young groups were combined) were calculated (note that in Fig. 7 the two groups can be distinguished) for each TDD separately. Correlations were generally low, although for the 200 and 1,000 ms TDDs they reached statistical significance. As can be seen from both Figs. 6 and 7, latency is lowest for the longest TDD, following the same general pattern observed for the correct responses. However, the peak error latency tended to be lower than the prosaccade latency from the calibration task; this is particularly marked for the 1,000 ms TDD (Fig. 7C).

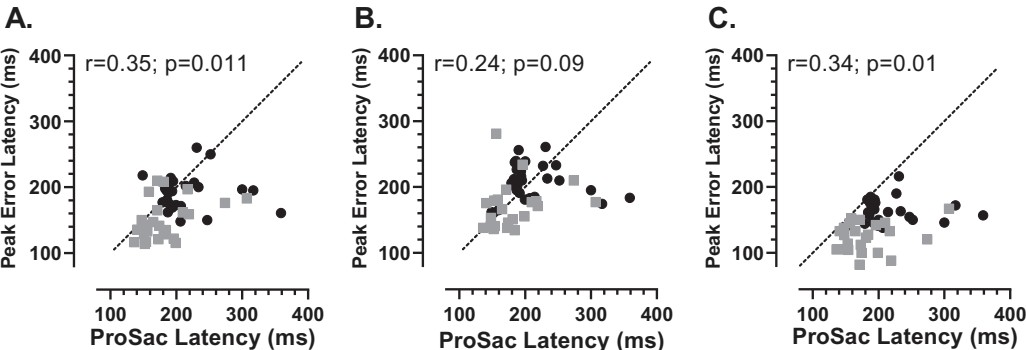

**Figure 7 Correlation between MDOR peak error and prosaccade latencies.** Correlation between MDOR error saccade latency for saccades occurring in the peak error range and median pro-saccade latency calculated from the calibration task. (A) TDD 200 ms; (B) TDD 600 ms; (C) TDD 1,000 ms. On each plot the Pearson correlation coefficient is shown (with accompanying p value), calculated for the whole dataset (i.e., old and young data combined), although on each plot the groups can be distinguished (old: black circles; young: grey squares). Dashed line shows the line of equality (x = y); points falling below this line indicate that the pro-saccade latency > MDOR error latency.

In order to assess whether the two participant groups might adopt fundamentally different strategies, sacrificing speed of response for more effective inhibition, we examined the data for evidence of speed/accuracy trade-offs (Fig. 8). Statistically there were no significant correlations between either $ER_t$ and latency in either group (young Fig. 8A; old Fig. 8C) or $ER_{pk}$ and latency (young Fig. 8B; old Fig. 8D). Thus there was no evidence of a difference in strategy in the two groups. However, there did appear to be a systematic difference in both groups between the shortest and longest TDDs. For the shortest TDD (200 ms) error rate varied widely over a relatively restricted latency range, whereas for the longest TDD (1,000 ms) latency appeared to be more variable.

## Comparison of original and modified MDOR tasks

We sub-divided the old group into those participants who were naïve to testing (N; $n = 16$) and those who participated in a previous experiment and returned for testing with the modified MDOR task (R; $n = 12$). These two groups were comparable in terms of age (N:65 ± 7 y; R:64 ± 6 y) and ACE III score (N:95 ± 3; R:96 ± 3). Both latency and total error rate were indistinguishable between these groups (Fig. 9).

There were a total of 24 participants (15 old, 9 young) for whom data were available both from the original MDOR task (MDOR-O) and the new modified task (MDOR-M). Of the 15 old participants, 12 completed MDOR-O first and MDOR-M second; only three completing the tasks in the opposite order. In the young group four did MDOR-O first. One of the key differences between the two versions of the task was that in MDOR-O there were only two TDDs (200 and 1,000), so these were the conditions for which comparisons could be made. As is clear from Fig. 10, both latency and total error rate were very similar in the two tasks whether considering data from all the participants in an omnibus analysis (Latency–Fig. 10A; Total Error Rate–Fig. 10D), or separately analysing data from young (Latency–Fig. 10B; Total error rate–Fig. 10F) and old (Latency–Fig. 10C;

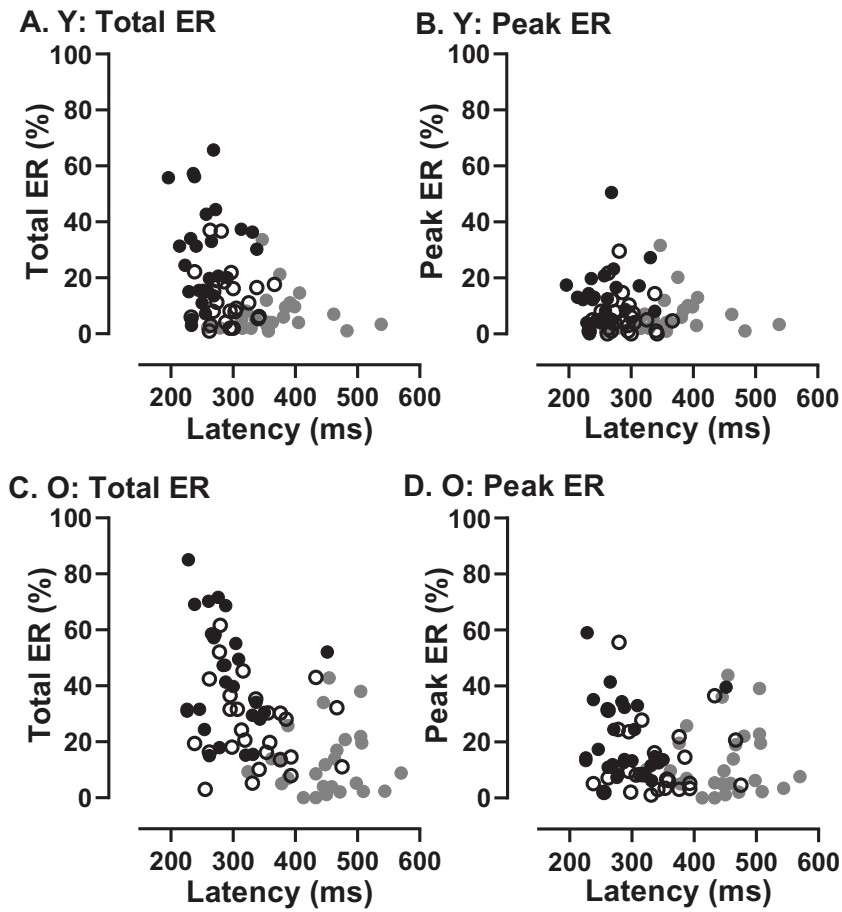

**Figure 8** **Relationship between error rates and latency of correct responses.** Plots of Error Rate (Total ER, $ER_t$: A, C; Peak ER, $ER_{pk}$: B, D) against latency for each participant in both young (A, B) and old (C, D) groups. In each plot, data from the three TDDs are distinguished: 200 ms gray symbols; 600 ms unfilled symbols; 1,000 ms black symbols.

Total error rate–Fig. 10E) groups. We subjected the latency and error rate data to separate ANOVA's of the same design, which treated TDD as a within and task (M vs. O) as between subjects factors. Unsurprisingly the task factor returned a statistically non-significant result for both latency ($F_{1,46}$ = 2.0, $p$ = 0.13) and error rate ($F_{1,46}$ = 0.12, $p$ = 0.74).

## DISCUSSION

Our objectives in the current experiment were to investigate the use of a modified MDOR task and further examine the effects of normal ageing that we had observed with the original task. The MDOR task was originally conceived to provide a means of investigating behavioural inhibitory control using eye movements (*Wolohan & Knox, 2014*). At that time, and subsequently (*Knox & Pasunuru, 2020*; *Knox, Heming De-Allie & Wolohan, 2018*; see also the Introduction) we discussed the advantages of the MDOR task compared to a number of alternatives. Essentially this is that it provides a less contaminated measure of the ability of participants to inhibit a prepotent response compared to those alternatives.

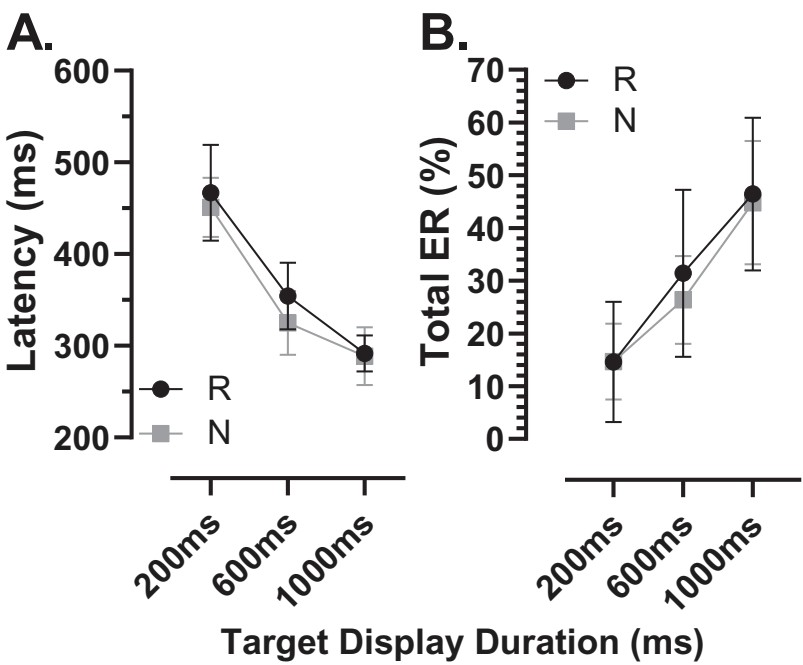

**Figure 9 Comparison in MDOR performance between naïve and experienced participants.** Comparison of performance in the modified MDOR task (MDOR-M) between those old participants who were naïve to testing (N) and those who had participated in a previous MDOR experiment with the original MDOR task (MDOR-O), and who had returned for testing (R). (A) Mean (±95% CI) raw latency for correct responses. (B) Total error rate. Full-size  DOI: 10.7717/peerj.11610/fig-9

Potentially this means that it might be a useful tool for detecting and investigating the breakdown in inhibitory control reported to occur in a range of conditions and pathologies (*Kaiser, Kuhlmann & Bosnjak, 2018*; *Manza et al., 2017*; *Rabi et al., 2020*; *Rochat et al., 2013*; *Smith et al., 2014*).

In the modified MDOR task, we increased spatial information available to participants and decreased temporal predictability compared to the original version, and provided better definition of trial start, end and the intertrial interval. One advantage flowing from these changes appears to be an improved yield of valid trials for analysis. In our previous experiment the mean yield per participant was 78% and 83% for young and old participants respectively (*Knox & Pasunuru, 2020*); here the comparable proportions were 97% and 93%. Participants reported that the modified task provided them with more confidence as to how long to fixate the target position towards the end of each trial; in the original version they were simply told to pause at the eccentric position. They also had more confidence about when they could blink because the intertrial interval was now clearly delineated. Much of our analysis focusses on error trials as these provide information specifically about inhibition failure. However, errors make up a minority of trials, and absolute numbers per participant can be small (particularly in younger participants). So task design features that increase the information available for analysis, without fundamentally altering task properties, are valuable.

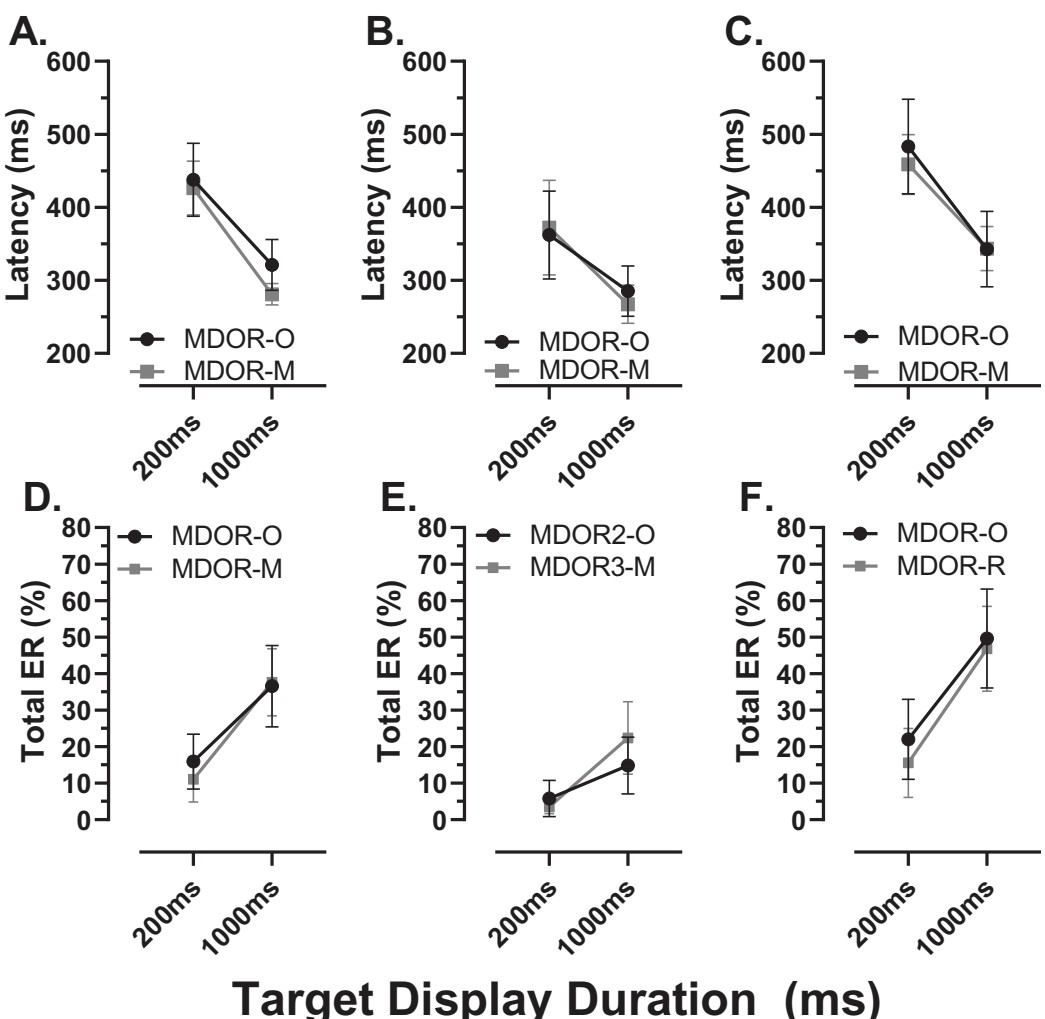

**Figure 10 Comparison in perfomance between the original and modified MDOR tasks.** Comparison of performance in the original MDOR task (MDOR-O) and the modified task described in the current study (MDOR-M), Mean (±95% CI) latency (A–C) and total error rate (D–E) is shown for all participants for whom data were available (A and D; *n* = 24) and then divided into young (B and E; *n* = 19) and old (C and F; *n* = 15) participants.

The general pattern of responses observed with the modified MDOR task was identical to that reported previously (*Knox & Pasunuru, 2020*; *Knox, Heming De-Allie & Wolohan, 2018*; *Wolohan & Knox, 2014*). Latency was related to TDD (longer latency at shorter TDDs), as was ER$_t$ (lower at shorter TDDs), and latency was longer and error rates higher in older compared to younger participants. As previously, most of the latency increase in the older group was due to general age-related slowing in the oculomotor system, replicating our previous result (*Knox & Pasunuru, 2020*). Prosaccade latency was obtained from the task used to calibrate eye tracker output rather than from a separate prosaccade task, in part to reduce the testing burden on participants. The calibration task is clearly different to the MDOR task, but it provided us with an estimate of prosaccade latency in a consistent manner for all participants. While a different prosaccade task might have

produced different absolute latencies, we would expect a similar latency difference between old and young groups. The difference we did observe (28 ms) was broadly consistent with (although smaller than) the difference we reported previously (47 ms), and consistent with other reports in the literature (*Eenshuistra, Ridderinkhof & Molen, 2004*; *Klein et al., 2000*). It provided us with a means of "correcting" for general age-related slowing, a procedure recommended in the literature (*Verhaeghen (2011)*, although see also *Ulrich, Mattes & Miller (1999)*). This is important because general slowing accounts for some of the age-related alterations in performance observed across tasks (*Salthouse, 1996*), particularly those associated with executive function (*Maldonado et al., 2020*). Notably after the latency "correction" a larger TDD-related modulation of latency still remained in the older group, but now measured over three TDDs rather than two (compare Fig. 3B in *Knox & Pasunuru (2020)* with Fig. 2B).

When the response to a target onset is successfully inhibited (so that there is no error response), the result is a correct MDOR response. We have argued that the latency of these correct MDOR responses reflects the level of inhibition around the time the target for the saccade is extinguished (the go signal). At a TDD of 200 ms, levels of inhibition at target offset are high, leading to long latencies; as TDD is increased the level of inhibition (at the time the correct saccade has to be initiated) falls, and so does latency. Our data suggest that older participants are capable of exerting high levels of inhibition initially. As can be seen from Fig 2B, at TDD = 200 ms the older group exhibited an additional latency increase, even once general age-related slowing was accounted for; the effect size ($d_s$) for the difference between groups at 200 ms in the corrected data was 0.83 and was statistically significant. At TDD = 600 ms this additional latency was greatly decreased (and was not statistically significant), and it was entirely absent at 1,000 ms.

The larger modulation in latency by TDD in the older group, a result we reported previously (*Knox & Pasunuru, 2020*), suggests that the maintenance of inhibition is more difficult for older participants. The magnitude of the modulation is very similar in both studies; previously comparing 200 and 1,000 ms TDDs the latency difference was 164 and 96 ms in old and young groups, here is was 168 and 113 ms. We might speculate that initially older participants are able to generate high levels of inhibition, perhaps even higher levels than young participants, as a strategy to compensate for poorer overall effectiveness of inhibition (reflected in generally higher error rates); this would produce the longer latency for correct responses at 200 ms TDD. However, an inability to maintain inhibition as TTD increased, would lead to latency dropping off, and a larger overall modulation.

Independent support for the general hypothesis that the MDOR task evokes inhibition, and that this is reflected in latency increases, is provided by the results of *Tari, Fadel & Heath (2019)* and *Tari & Heath (2019)*. They used the MDOR task in a task switching paradigm and found that prosaccades following an MDOR task had increased latency. As with task switching costs observed with mixed AS and prosaccade tasks, they interpreted this as being due to the effects MDOR-induced inhibition.

While the latency of correct responses may reflect levels of inhibition, error rates reveal its effectiveness; these were consistently higher in the old compared to the young group.

However, the distributions (Fig. 4) indicate that it is important to distinguish between different types of error. We have previously discussed the advantages of average, compared to pooled, distributions for prosaccade (*Knox & Wolohan, 2014*; *Knox, Wolohan & Helmy, 2017*), antisaccades (*Knox et al., 2012*) and MDOR tasks (*Knox & Pasunuru, 2020*; *Wolohan & Knox, 2014*). Here we have used them as a means of graphically summarising responses across groups, in a form which allows the identification of significant features. For errors, the clearest feature is the peak occurring just after target onset, which we used to derive the peak error rate ($ER_{pk}$). We suggest that this provides a direct index of inhibitory failure as given their timing, the errors which make up this peak are most likely uninhibited responses to target onsets. Note that a similar claim might be made for antisaccade errors (i.e., that AS errors are uninhibited prosaccades). However, as outlined in the Introduction, this interpretation is complicated by there being two processes competing for behavioural expression in the AS task. An AS error therefore is not necessary the result of a failure of inhibition. In the MDOR task there is no such competition.

However, if early MDOR errors were only uninhibited responses to the target onset, then there is no reason to expect $ER_{pk}$ to differ with TDD. It is striking that in both participant groups $ER_{pk}$ is noticeably raised in the 1,000 ms condition (Figs. 4C, 4F; see also Fig. 5A) compared to the two other TDDs where indeed it is similar. It seems unlikely that this is a chance occurrence, particularly as it was observed in the distributions in our previous study in both young and old groups (*Knox & Pasunuru, 2020*; Figs. 5B, 5D). This might suggest some other process influencing $ER_{pk}$ at least when TDD is long. However, supporting the idea that these errors are responses to target onsets (and are therefore for the most part inhibitory failures), the latency difference between groups was similar to that observed between the prosaccade and MDOR correct response latencies (Fig. 6), and there was a (modest) correlation between the peak error and prosaccade latency across participants (Fig. 7).

The distributions also provide further information about the dynamics of inhibition and its difference between groups. Errors continued (at a low rate) throughout the period when central fixation was required, up to the target offset. But for the 1,000 ms TDD (Figs. 4C, 4F), the error rate rose slowly as the fixation period wore on, particularly from around −400 ms (600 ms after the target onset). With TDD = 600 ms (Figs. 4B, 4E) the error rate was flat in the young group and falling in the old group. The residual error rate ($ER_{r}$; Fig. 5B), differed for 600ms TDD but not 1,000 ms TDD, consistent with the young participants being able to maintain inhibition at a higher level for longer than the older group. The absolute number of errors at these timings is small, so these patterns should be interpreted cautiously. But perhaps they provide evidence that the increased inhibition initially evoked in the MDOR task at target onset can only be maintained for a few hundred milliseconds, and that the ability to maintain inhibition declines as we age.

If essentially the same processes (or their failure) accounted for both the error rates and the latency of correct responses, then we would expect to see a relationship between them. A participant who was able to exert high levels of inhibitory control might have a low error rate (particularly a low $ER_{pk}$) and a longer latency for correct responses.

There was no clear evidence of this type of relationship across conditions and groups (Fig. 8). What the data in Fig. 8 do suggest is a difference between the shortest and longest TDDs, but with a similar pattern in both groups. At TDD = 1,000 ms for both $ER_t$ and $ER_{pk}$, the error rate was highly variable between participants while latency appeared to be less variable. The opposite pattern is seen for TDD = 200 ms. Latency is longer and error rate higher, but the relationship between them is altered; latency appears to be more variable and error rate less variable. As the same general pattern was seen in both old and young groups, these data provide no evidence of a different control strategy between groups.

Among the older participants, we had comparable groups of naïve and returning participants. The experience of having visited the lab and completed the original MDOR task did not influence performance in the modified MDOR task. While there was a period of several months between tests on the two different versions of the task, there was little evidence of learning or exposure effects. Procedures in the lab were of course designed to produce exactly this result by carefully explaining and demonstrating tasks and providing practice trials before data collection began with the aim of achieving stable performance in any one session of testing. Such stability in performance over time does appear to be a general feature of at least simple saccade tasks. In the absence of training (exposure to large numbers of trials over a relatively short period of time), this has been taken to support the concept of an oculomotor phenotype (*Bargary et al., 2017*; *Knox & Wolohan, 2015*). Our study was not designed to investigate the stability of MDOR performance over time; longitudinal testing will be needed to demonstrate that MDOR performance provides a means of measuring trait inhibitory control as opposed to providing a means of measuring the inhibitory state at the time of testing.

As demonstrated in Fig. 10 there was little evidence of difference in latency and total error rate either overall (Figs. 10A, 10D) or in either the young (Figs. 10B, 10E) or old (Figs. 10C, 10F) groups when we compared the original and modified MDOR tasks in the same participants. This suggests a degree of stability across versions of the MDOR task and perhaps also a lack of sensitivity of the measured parameters to task context. One concern with the original MDOR task might have been that with only two TDDs (200 and 1,000 ms), participants might have been able to predict the time of target offset, generally supressing latency. The addition a the third TDD would be expected to make this less likely. If such an effect were present, then with the modified task a general increase in latency might be expected. However, there is no consistent evidence that this is the case. Using task design to vary performance is, of course, occasionally useful depending on the type of participants being tested. In the AS task, the use of gap AS tasks tends to increase error rates, usefully preventing floor effects in healthy control groups in clinical studies. We have shown previously that gap and overlap effects do not influence MDOR performance (*Knox, Heming De-Allie & Wolohan, 2018*). So too the modifications made in the current study had little effect on performance.

As is well recognised, inhibitory control is not a monolithic construct (*Aron, 2011*), several taxonomies have been used to describe it (*Rey-Mermet, Gade & Oberauer, 2018*)

and many tasks have been used to investigate it. Correlations between tasks tend to be poor (*Friedman & Miyake, 2004*; *Rey-Mermet, Gade & Oberauer, 2018*) and even tasks that might be thought of as being relatively similar (eg manual go/no go and SSRT; AS and MDOR task) may actually involve different mechanisms or processes (*Raud et al., 2020*; *Wolohan & Knox, 2014*). The existence and nature of an age-related inhibitory control deficit has been a matter of recent debate (*Rey-Mermet & Gade, 2017*; *Verhaeghen, 2011*). However, the MDOR task does appear to offer insights into the effectiveness of inhibition and perhaps also its dynamics, at least with respect to oculomotor control. Using the modified MDOR task in the current study, we have confirmed that the effectiveness of inhibitory control declines in healthy ageing. We suggest that the MDOR task (in either original or modified versions) provides a useful means of investigating multiple aspects of inhibitory control.

## ACKNOWLEDGEMENTS

We are grateful to all the participants who took part in these experiments.

### Funding

This study was supported by a visiting scholarship to Dongmei Liang from the China Scholarship Council (#201906755035). The funders had no role in study design, data collection and analysis, decision to publish, or preparation of the manuscript.

### Grant Disclosures

The following grant information was disclosed by the authors:
Dongmei Liang from the China Scholarship Council: #201906755035.

### Competing Interests

The authors declare that they have no competing interests.

### Author Contributions

- Paul C. Knox conceived and designed the experiments, performed the experiments, analyzed the data, prepared figures and/or tables, authored or reviewed drafts of the paper, and approved the final draft.
- Dongmei Liang performed the experiments, analyzed the data, authored or reviewed drafts of the paper, and approved the final draft.

### Human Ethics

The following information was supplied relating to ethical approvals (i.e., approving body and any reference numbers):

This study was conducted under an ethical approval from the University of Liverpool Research Ethics committee (Reference No. 2933).

## Data Availability

The data are available at figshare: Knox, Paul (2021): Modified MDOR task-effect of normal ageing on performance. figshare. Dataset. DOI 10.6084/m9.figshare.14061053.v1.

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
