# Peer review of "Confirmation of age-related alterations in inhibitory control using a modified minimally delayed oculomotor response (MDOR) task"

_PeerJ, doi:10.7717/peerj.11610_

## Round 0.1 · original submission · Minor Revisions

Your manuscript has now been seen by two reviewers. You will see from their comments below that they find your work of interest, and some constructive points are worth considering. We, therefore, invite you to revise and resubmit your manuscript, taking into account the points raised. Please highlight all changes in a separate tracked-changes manuscript text file.

·

Basic reporting

The manuscript is well written and demonstrates the requisite level of expertise and familiarity with the extant oculomotor inhibition literature. The structure of the manuscript appears to conform to PeerJ standards and figures are relevant, high quality, well labelled and described. In terms of raw data, I did not view a statemen indicated that values were available in a public archive.

Experimental design

I have a number of questions, comments, concerns about the methods and statistics and present them in detail below.
• Line 117: The authors indicate that their previous work showed that the general difference between young and older adults was attributed to a “general age-related slowing in the visuomotor system”. Having read through the manuscript I now understand this was done by subtracting saccades initiated at target onset with MDOR saccades. That said, a reference at page 117 to Salthouse (1996: Psychol Rev) would be important, as would a description of how slowing with age can be attributed to hardware and/or software changes.
• Line 123: The rationale for the study is – in part – attributed to the fact that the authors’ previous work (which did not employ placeholders) was associated with increased spatial uncertainty owing to the lack of extraretinal feedback. This might be true; however, the authors could have examined the issue by contrasting the variability of endpoints and peak saccade velocity given that the metics represent hallmark features of spatial uncertainty (Edelman and Goldberg 2001: J Neurophsiol).
• Line 136. It is not clear why the study employed the same participants from previous work as well as naïve participants. This rationale needs to be clearly and carefully made.
• Line 142. Was a pharmaceutical screening tool used for older (and younger adults)? This is not a trivial question as an extensive list of pharmaceuticals commonly used in the older adult population decrease oculomotor/information processing speed (for review see, Reilly et a. 2008: Brain Cogn).
• Lines 173-174. What does “[…] stepping through it […]” mean?
• Line 219. Persons in the age group with a mean age of 65 years are referred to as the “old” group. The term “older” group is more appropriate.
• Line 220. Not all readers will be familiar with the ACE III (MMSE and MoCA are more common) and as such please indicate that all participants scored in the normal range (i.e., did not have clinical or preclinical cognitive decline/disease).
• Lines 228-245. This section is overly dense and requires simplification. The section simply needs to state that main effects were observed for group and TDD and their interaction and that in general RTs were shorter for the younger than older group at the 200 and 600 – but not 1000 ms – TDD. The between-group comparison is more important than the within-group contrast of TDD (i.e., the decrease in RT with increasing TDD can be inferred from Figure 2). Moreover, effect size measures are reported for between-group comparison but not p-values, whereas p-values and not effect sizes are reported for the within-group comparison of TDD. Please be consistent.
• Related to the above, Figure 2a shows overlap between error bars for young and older adults at the 1000 TDD and the page 244 of the manuscript indicates an effect size for this comparison of 0.78. If I work backwards from the effect size and compute a t-ratio I find a value of 2.88 which would be associated with a p value <.01. So, it would appear to be a reliable effect based on the effect size; however, the overlap between errors bars by definition demonstrates that the two groups do not reliably differ. Hence, the authors need to clearly articulate whether groups reliably differed at each TDD.
• Related to the above, the same issues apply to Figure 2b and the statistics presented for the corrected RTs.
• I really like the subtractive technique employed by the authors. I do, however, think It fair that a revision should note that potential pitfall(s) of this Donder’s-based approach (Ulrich et al. Acta Psychologica: 1999).
• Lines 261-271. A regression approach, and the use of slopes, to contrast RT and TDD scaling between-groups is interesting. I would, however, caution using the approach as it is does not appreciably add to the information presented in the previous two paragraphs and does not contain the experimentwise error rate.
• Line 280. The results for total error yielded an interaction between group and TDD and the authors report the effect sizes for each between-group comparison of TDD. The effect size values appear sufficient magnitude to establish a between-group difference at each TDD; however, the Figure 3 shows that error bars for the 200 TDD overlapped between groups. Hence, did groups differ at this TDD. This result needs to be clearly articulated.
• I liked the analyses associated with Figure 4 and appreciate that the associated histograms provided a meaningful overview of the RT and error data.
• The Discussion is well-written. If I had one complaint it is that it was very long. Of course, the authors’ depth of interpretation stands as their own and I leave it to them as to whether they should abbreviate the Discussion.

Validity of the findings

My opinion is that the validity of findings can be rated as "very high".

·

Basic reporting

The authors presented pretty clear evidence in the introduction that the MDOR task is a better task for studying inhibition compared to antisaccade and memory-dependent saccade tasks. However, I think the amount of focus on comparing the modified and original versions detracts from the investigation of the effects of aging on inhibition. In particular, I think the comparison of the two versions could be dropped since the sample size for participants who completed both is pretty small.

Experimental design

The task schematic and timeline are clear and informative. I do greatly prefer the figures to be embedded in the body of the manuscript, but this might be in line with journal instructions. The distribution plots are useful for visualizing the erroneous and correct responses.

The analysis is very thorough and considers multiple facets of behavior including peak error rate, residual error rate, latencies, and the relationship between errors and prosaccades. Could you just explain how the residual error rate was determined more clearly? I was a bit confused how the bins between -250 and 0 were used. Relatedly, why wasn’t a larger time range used for the long TDD?

Validity of the findings

No comment

Additional comments

In this manuscript, Knox & Liang used a modified minimally delayed oculomotor response (MDOR) task to investigate response inhibition in young and older adults. Older adults made more errors and were slower compared to young adults. An examination of latency distributions suggested that older adults were capable of successfully inhibiting responses but couldn’t maintain the level of control as well as younger adults. The study is well designed and the analyses were thorough. I have some relatively minor suggestions for improvement, but overall saw no major issues with the manuscript.

The authors presented pretty clear evidence in the introduction that the MDOR task is a better task for studying inhibition compared to antisaccade and memory-dependent saccade tasks. However, I think the amount of focus on comparing the modified and original versions detracts from the investigation of the effects of aging on inhibition. In particular, I think the comparison of the two versions could be dropped since the sample size for participants who completed both is pretty small.

The task schematic and timeline are clear and informative. I do greatly prefer the figures to be embedded in the body of the manuscript, but this might be in line with journal instructions. The distribution plots are useful for visualizing the erroneous and correct responses.

The analysis is very thorough and considers multiple facets of behavior including peak error rate, residual error rate, latencies, and the relationship between errors and prosaccades. Could you just explain how the residual error rate was determined more clearly? I was a bit confused how the bins between -250 and 0 were used. Relatedly, why wasn’t a larger time range used for the long TDD?

---

## Round 0.2 · accepted · Accept

Thank you for the revised manuscript and response letter. I am pleased to inform you that your manuscript has been accepted for publication. However, before publication, please take into account the correction suggested by Reviewer 1.

·

Basic reporting

Acceptable.

Experimental design

Acceptable.

Validity of the findings

Acceptable.

Additional comments

I found the paper interesting and appreciate the attention to data analyses. I also found that the authors provided an acceptable revision. The only comment I reserve is that the term "old" not be used to describe older adults. This recommendation is based on APA and APS standards.

·

Basic reporting

no comment

Experimental design

no comment

Validity of the findings

no comment

Additional comments

no comment - my questions were addressed